# Greenness Indices from a Low-Cost UAV Imagery as Tools for Monitoring Post-Fire Forest Recovery

## Asier R. Larrinaga [1,2,3,*] and Lluis Brotons [1,4,5]

1   InForest Joint Research Unit, (CTFC-CREAF), 25280 Solsona, Spain; lluis.brotons@ctfc.cat
2   eNeBaDa, Santiago de Compostela, 15892 A Coruña, Spain
3   Forest Genetics and Ecology Group, Biologic Mission of Galicia (CSIC), 36413 Pontevedra, Spain
4   CREAF, 08193 Cerdanyola del Vallès, Spain
5   CSIC, 08193 Cerdanyola del Vallès, Spain
*   Correspondence: asier@enebada.eu or arodriguez@mbg.csic.es; Tel.: +34-659-087-889

**Abstract:** During recent years unmanned aerial vehicles (UAVs) have been increasingly used for research and application in both agriculture and forestry. Nevertheless, most of this work has been devoted to improving accuracy and explanatory power, often at the cost of usability and affordability. We tested a low-cost UAV and a simple workflow to apply four different greenness indices to the monitoring of pine (*Pinus sylvestris* and *P. nigra*) post-fire regeneration in a Mediterranean forest. We selected two sites and measured all pines within a pre-selected plot. Winter flights were carried out at each of the sites, at two flight heights (50 and 120 m). Automatically normalized images entered an structure from motion (SfM) based photogrammetric software for restitution, and the obtained point cloud and orthomosaic processed to get a canopy height model and four different greenness indices. The sum of pine diameter at breast height (DBH) was regressed on summary statistics of greenness indices and the canopy height model. Excess green index (ExGI) and green chromatic coordinate (GCC) index outperformed the visible atmospherically resistant index (VARI) and green red vegetation index (GRVI) in estimating pine DBH, while canopy height slightly improved the models. Flight height did not severely affect model performance. Our results show that low cost UAVs may improve forest monitoring after disturbance, even in those habitats and situations where resource limitation is an issue.

**Keywords:** low-cost UAV; greenness index; *Pinus nigra*; *Pinus sylvestris*; forest regeneration; flight altitude; small UAV

## 1. Introduction

During recent years, UAVs (unmanned aerial vehicles) have grown increasingly popular for the study of land and its cover [1,2]. This trend is the consequence of a recent exponential development of both the UAV industry and the do-it-yourself community, fostered by the technological advances in robotics and the miniaturization of electronics.

Forest research is one of the fields where the use of UAV has promised immediate benefits [1]. UAVs allow the reduction of costs of airborne photography and LIDAR, and approach technology to its final user [2,3]. By doing so, it offers great flexibility. UAV deployment is fast and cheap, ensuring rapid responses to the needs of both academia and industry while allowing for repeated sampling with no limits on deployment periodicity. A new, tailor-cut telemetry sampling strategy is now possible, designed to fit the specific needs of each case study [3].

UAVs have been used to discriminate among species and provide estimates of tree and stand size, tree cover, canopy height, gap abundance or even productivity, alone or in combination with LIDAR

data [4–12]. Tree health and pathogen or parasite attack have also been evaluated by means of UAV telemetry [13].

Therefore, research on different methodologies and, more specifically, on the accuracy of those data estimated by means of UAV imagery has bloomed during the last five years [2]. The accuracy level of these kind of works is continuously increasing and it is a major focus of an important body of research on UAV use in forestry [2,4,14–16]. Spatial accuracy and automatic tree detection in particular are rapidly improving [9,12,16–19].

Nevertheless, accuracy is not always the main constraint to UAV use in forestry science. In fact, pursuing high standards of spatial and analytical accuracy is a time consuming goal that often requires major investment [20]. While this might be a sensible approach when working with economically exploited forests and woodlands, it might hinder the development of forestry research in other areas or research fields, such as disturbance response, where immediate revenues cannot be envisaged. In these situations, the absence of any kind of data is common while economic resources and work forces are scarce. A reduction of both accuracy and the cost of deployment could hence help to get general data on the ecology of the forest that would greatly improve our knowledge.

In this work we explore the use of a low-cost UAV platform as a tool for monitoring recovery of a Mediterranean forest after a strong disturbance. Our objective is to assess post-fire pine regeneration (Scots pine, *Pinus sylvestris*, and black pine, *P. nigra*) in an area affected by a wildfire where oak has become dominant. In order to identify pine cover in our area, we compare the use of four different greenness indices at two different flight heights, recorded areas and hence costs. Low-cost UAV platforms offer several advantages for this task: (1) they are affordable and easy to use for stakeholders and practitioners, (2) they can be controlled and analyzed with free photogrammetric software and 3D reconstruction web services, (3) due to their small size they can be flown even in remote areas, where access by vehicle is difficult, (4) they can be flown on demand at no cost, allowing one to choose the flying time depending on weather, plant phenology or other logistical constraints, and (5) they offer ultra-high resolution, allowing one to detect pine trees even at the early stages of regeneration. In this context, we aimed to develop a tool to monitor the emergence of pines that grow among the oaks and test the suitability of low-cost UAVs as a cost effective monitoring tool.

## 2. Materials and Methods

We carried out our work in the municipality of Riner, in the Lleida province (Catalonia, Spain), where an extensive wildfire burnt down around 25,000 ha of pine-dominated woodland in 1998. Most of the area have apparently recovered to a great extent in these last 20 years. A closer look, however, depicts a different picture. The effect of wildfires might have drawn the forest beyond its resilience threshold, causing a change to a new alternative equilibrium state [21]. In fact, although tree cover seems to be almost completely recovered, its species composition has changed in a radical way, as Portuguese oak (*Quercus faginea*) thrives in the burnt land where pines (*P. nigra* and *P. sylvestris*) were dominant before the wildfire.

Our two sites, La Carral and Cal Rovira-Sanca, are 3.7 km away from each other and both of them supported similar Mediterranean forests before the 1998 fire, on marl, limestone, and sandstone rocks (Figures S1 from reference [22]).

### 2.1. UAV Deployment and Field Sampling

We carried out two flights in each of the sites, one at a height of 50 m and a second one at a height of 120 m over terrain, by means of a DJI Phantom 2 quadcopter. We flew at two altitudes with the aim of exploring the effect of flight height on image quality and subsequent capacity to characterize pine tree recovery. The copter was equipped with a Phantom Vision FC200 camera, manufactured by DJI, which has a resolution of 14 Mpx with a sensor size of 1/2.3″ (6.17 mm * 4.55 mm), a focal length of 5 mm and an electronic rolling shutter [23]. We set the camera to shoot one picture every three (La Carral) and five (Cal Rovira-Sanca; Table 1) seconds with an automatic exposition mode setting (with ISO 100).

All four flights were carried out in March 2015 (17 years after the fire) under optimal weather conditions (sunny days with low wind intensity); in Cal Rovira-Sanca in the morning and in the afternoon in La Carral (Table 1). By flying in winter, we ensured a good spectral discrimination between pines (the only perennial tree species group in the area)—and the remaining components of the canopy, namely Portuguese oaks, as the latter still hold their dry leaves on the branches and do not shed new leaves until spring.

In each of the sites we selected a sampling area near the center of the flight zone and identified all the pines growing there (Figure 1). Each pine was geo-referenced and identified to the species level. In addition, height and diameter at breast height was measured (DBH) for each individual. DBH was measured with the aid of a measuring tape with a 1 mm resolution. Tree height was measured with the same measuring tape, except for the highest trees, where a measuring pole was used (1 cm resolution). We measured DBH at 1 m height for logistical reasons, given the abundance of low pine trees.

**Table 1.** Characteristics of the four flights carried out in La Carral and Cal Rovira-Sanca. Time refers to mean time of each flight. Centre of scene: geographic center of each scene in UTM, fuse 31, datum ETRS89. Flight height gives nominal values. Area: coverage area of each flight. Pixel size: size of ground pixel. Reprojection error: difference between a point in an image and its position according to the fitted 3D model. Motion blur: blur due to linear movement (rotation effects are not included).

| Site | La Carral | La Carral | Cal Rovira-Sanca | Cal Rovira-Sanca |
|---|---|---|---|---|
| Date (DD/MM/YY) | 03/07/18 | 03/07/18 | 03/03/18 | 03/03/18 |
| Time (UTC) | 16:59 | 17:26 | 10:13 | 10:25 |
| Sun elevation angle (°) | 19.08 | 14.49 | 27.29 | 28.96 |
| Sun azimuth angle (°) | 243.95 | 249.07 | 130.07 | 132.86 |
| Centre of Scene (UTM31N-ETRS89) | (378698,4640314) | (378715,4640324) | (375511,4642336) | (375522,4642349) |
| # of images | 160 | 147 | 90 | 67 |
| Flight height (m) | 50 | 120 | 50 | 120 |
| Flight speed (m/s) | 4 | 4 | 4 | 4 |
| Area (ha) | 5.82 | 24.6 | 7.54 | 21.3 |
| Side overlap (%) | 55 | 65 | 48 | 62 |
| Forward overlap (%) | 74 | 89 | 57 | 82 |
| Effective overlap (# image/pixel) | 3.40 | 7.94 | 2.88 | 4.80 |
| Pixel size (cm) | 1.46 | 4 | 1.59 | 3.96 |
| Reprojection error (pixel) | 8.31 | 7.95 | 1.78 | 1.76 |
| Mean shutter speed (s) | 1/288 | 1/312 | 1/457 | 1/525 |
| Motion blur (cm - pixel) | 1.39–0.95 | 1.28–0.32 | 0.88–0.55 | 0.76–0.16 |

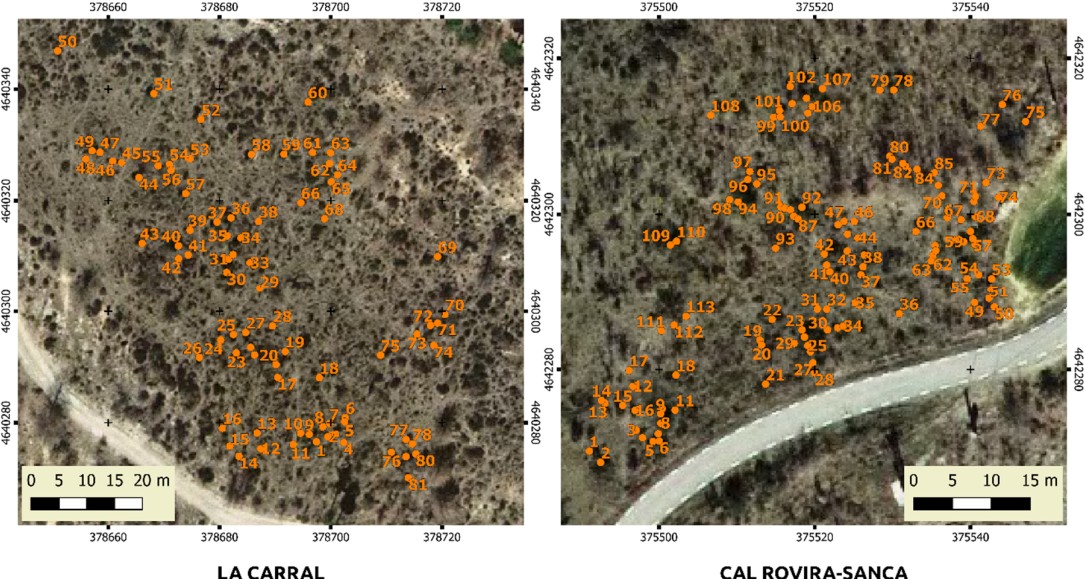

**Figure 1.** Aerial ortophotographs of both sites, with overlapped positions of all the pines found in the sampling areas.

## 2.2. Image Analysis

We aimed to get an easy workflow and to that end, we tried to minimize parameter tweaking during the whole process. Hence, most of the procedures were carried out using automatic tools and algorithms, as well as default values for the various parameters. For the rest of the section, we will be giving details on the parameters we set specifically, while omitting those parameters that were set to their default values.

First, we adjusted all the aerial images with an image managing and editing software [24], by sequentially applying the Automatic Levels and Automatic Contrast tools. By doing so we corrected some inequalities in color balance produced by the FC200 camera and improved their contrast for further analysis.

We carried out 3D reconstruction within an advanced terrain-oriented software that creates 3D models, point clouds and orthomosaics from a set of images on the same subject, by means of SfM (structure from motion) techniques [16,25,26]. We followed the general workflow suggested in its user manual [27]. Photo aligning and estimation of camera locations and calibration parameters were carried out as a first step, together with the building of a sparse point cloud (Figure 2; Figure S2). This sparse point cloud was then used to create a dense point cloud with mean point density of 407 points/m for the 50 m high flights and 69 points/m for the 120 m high flights. Finally, a digital elevation model and an orthomosaic were built from this point cloud, using the corresponding default parameters in our 3D reconstruction software [25].

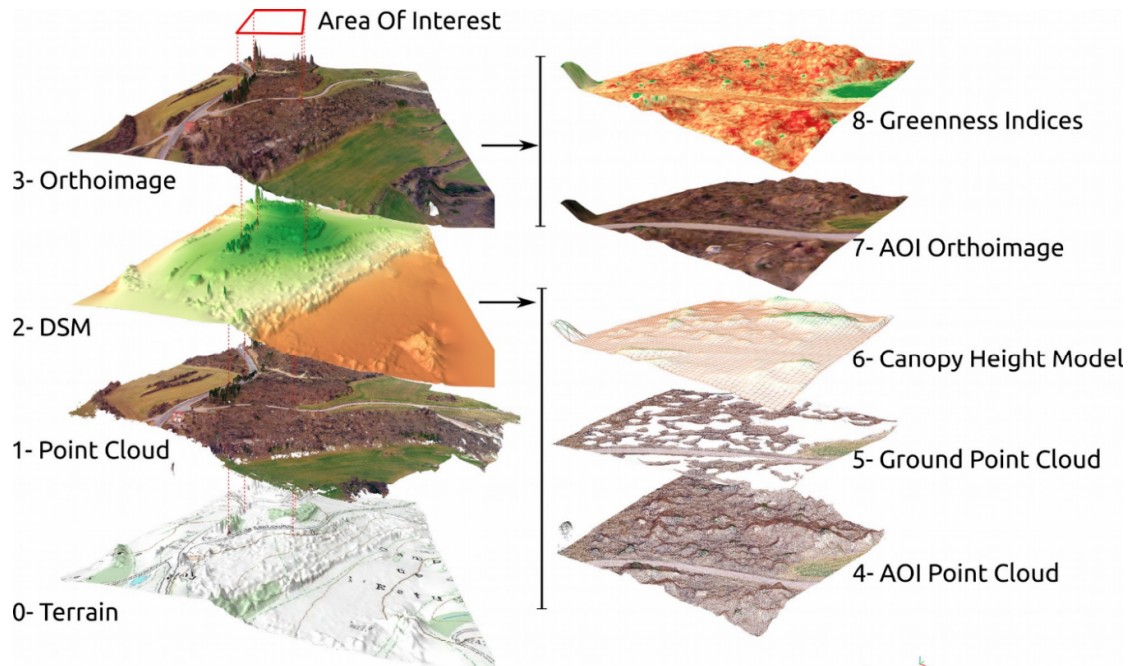

**Figure 2.** General workflow for the analysis of UAV imagery, as shown by its intermediate output images, point clouds and 3D models. A detailed workflow is provided as Figure S2-1. Orthoimages are shown draped on the obtained DSM, in order to improve figure display.

As a reference image to set control points, we used the most recent orthophoto of the area from the National Plan of Aerial Orthophotography (Plan Nacional de Ortofotografía Aerea, PNOA), which has a pixel resolution of 25 cm. Altitude of control points was manually extracted from the official digital terrain model (DTM05, 5 m spatial resolution) released by PNOA. We set between 7 and 9 control points per flight by selecting particular features that could be identified both in the orthomosaic obtained from the 3D reconstruction software [25] and in our reference aerial photograph, such as stones, rocks, fallen trunks and artificial features (corners, road paint, etc.). We then used those control

points to reconfigure the cameras and rebuild the dense point cloud, the DSM (digital surface model) and the orthomosaic.

The entire process within the 3D reconstruction software was carried out with by-default parameters and pixel size was only set during the export step. We exported 50 m high flight orthomosaics and DSM with a pixel size of 2 and 8 cm, respectively and those from 120 m high flights with pixels of 4 and 16 cm.

We clipped the obtained point cloud by means of a specific free tool for forest LIDAR data analysis [28]). Misplaced points were detected in those clipped point clouds obtained from 50 m high flights and hence, they were cleaned up with a point cloud editing and registering open software [29]. Those from both 120 m high flights did not need such a cleaning process. We extracted those points corresponding to the ground level, by means of a filtering algorithm adapted from Kraus and Pfeifer [30] and then computed a digital terrain model (DTM) by using the mean elevation of all ground points within a cell [28]. From the highest elevation point in each cell and the created DSM, we built a canopy height model with a cell size of $10*10$ cm$^2$ [28].

We calculated four different greenness indices from the obtained orthomosaic: excess green index (ExGI), green chromatic coordinate (GCC), green-red vegetation index (GRVI) and visible atmospherically resistant index (VARI). We hereby provide the rationale for their selection and their definition:

- The excess green index (ExGI) is a contrast index that has been shown to outperform other indices in discriminating vegetation [31,32] and is one of the most widely used indices in the visual spectrum. It is defined as:

$$ExGI = 2*G - (R+B) \tag{1}$$

- The green chromatic coordinate (GCC) has also been used to detect vegetation and analyze plant phenology and dynamics [31,32]. Both ExGI and GRVI correlate with measurements made with a SpectroSense narrow spectrometer [33], but GRVI is far less sensitive to changes in scene illumination [32]. It is simply the chromatic coordinate of the green channel expressed as a proportion of the sum of coordinates:

$$GCC = \frac{G}{R+G+B} \tag{2}$$

- The green red vegetation index (GRVI) was first used by Rouse et al. [34] who concluded it could be used for several measures of crops and rangelands. Their conclusions have been later confirmed in several occasions [35–38]. It responds to leave senescence of deciduous forests in a parallel way to that of NDVI [37] and hence could be useful for discriminating senescent leaves from green needles. This index is given by:

$$GRVI = \frac{G-R}{G+R} \tag{3}$$

- Lastly, the visible atmospherically resistant index (VARI) was proposed by Gitelson et al. [39]. It is an improvement of GRVI that reduces atmospheric effects. Although this is not an expected severe effect in low flying UAV platforms, it might locally be so, at Mediterranean sites with large amounts of bare soil. In addition, it has been reported to correlate better than GRVI with vegetation fraction [39]. It is defined as:

$$VARI = \frac{G-R}{G+R-B} \tag{4}$$

We calculated all four greenness indices directly from their digital numbers (DN) as provided by the JPEG format provided by the camera, instead of calculating reflectance values. JPEG compression

is a "lossy" compression method that prioritizes brightness over color, resulting in an important reduction of dynamic range of the picture and certain degree of image posterization (hard color changes and lower number of colors in the image). As a result, radiometric resolution decreases, which should reduce the ability to discriminate among different terrain or vegetation categories based on their visual spectrum. However, JPEG images still can successfully discriminate among different phenological stages of the vegetation, except at the most extreme compression ratios (97%) [32].

As we were not using reflectance to calculate these indices, their properties might not be the same as those described by other authors [35,37,39]. Particularly, our calculated indices cannot be directly compared with indices from different studies or even different flights, as they are sensitive to sensor characteristics and scene illumination. Still, we expected them to be useful in our context and decided to use this simpler approach, as our aim was to get as simple a workflow as possible, in order to allow for an easy, handy use of UAV imagery by non-expert users. Similar approaches have been successful in the past, even when analyzing repeated images over time [32,33,40,41]. Despite important effects of camera model and scene illumination in absolute greenness indices, the changes in plant phenology (changes from green to senescent leaves) were correctly detected by using uncorrected DNs [41].

Index calculations and their posterior analysis were carried out in an open SIG software [42], by combining the use of raster calculator with specific tools of zone statistics and spatial joining. The whole process for each index, after its calculation, involved the following steps:

- Applying the greenness threshold to the indices layers, in order to erase all non-green pixels, which were set to 0, For ExGI, GRVI and VARI, we defined green pixels as those with values greater than 0 and reclassified values less than 0 as 0. For GCC, the applied threshold was 1/3, and hence all values equal to or lesser than 1/3 were set to 0.
- Calculating the zone statistics of the greenness index and the filtered canopy height model for each 5*5 m$^2$ cell within the study area. Zone statistics produces six different measures of the index value per each cell: count, mean, standard deviation, median, maximum and minimum.
- Calculating the pooled DBH of all measured pines within the cells of this same grid.

### 2.3. Statistical Analysis

We aimed to assess the recovery of pines in the areas burnt in 1998. Hence, we selected the sum of the diameter at the breast height (DBH) of all pines within a cell as our response variable (Figure 3). DBH was highly correlated to pine height (see Supplementary Material S1) and its measure in pines is easier and less prone to measurement errors. DBH has also been related to many other morphological and functional traits [43–49] and hence it is open to a more insightful analysis. The sum of DBH values from all the pines in a grid cell combines the effect of density (number of pine trees per cell) with that of tree size (DBH). Given the relationship of DBH with crown size and foliage area [50], the sum of DBH is expected to correlate also with canopy cover [51]. In fact, basal area and tree density can successfully be used to predict canopy cover in pines [52–54].

First, we carried out simple linear regression analysis of the sum of DBH on the four greenness indices and on the canopy height model. For each greenness index and the canopy height model, we fitted six models, where the derived explanatory variables corresponded to six different summary statistics per 5 m grid-cell: the count of points with non-zero values for the corresponding index or canopy height model and the sum, mean, standard deviation, maximum and median of the values of all points within the cell. Our aim was to explore which statistic could be better suited for estimating sum of DBH from greenness index and canopy height maps. Then we tried to improve model fit by combining greenness indices with pine canopy height, by means of multiple linear regression analysis.

$R^2$ (for simple linear models) and adjusted $R^2$ (for multiple linear models) and scatterplots were used throughout the process to assess model fit and comparing models.

All statistics were carried out in R [55], by means of the R-Studio integrated development environment [56].

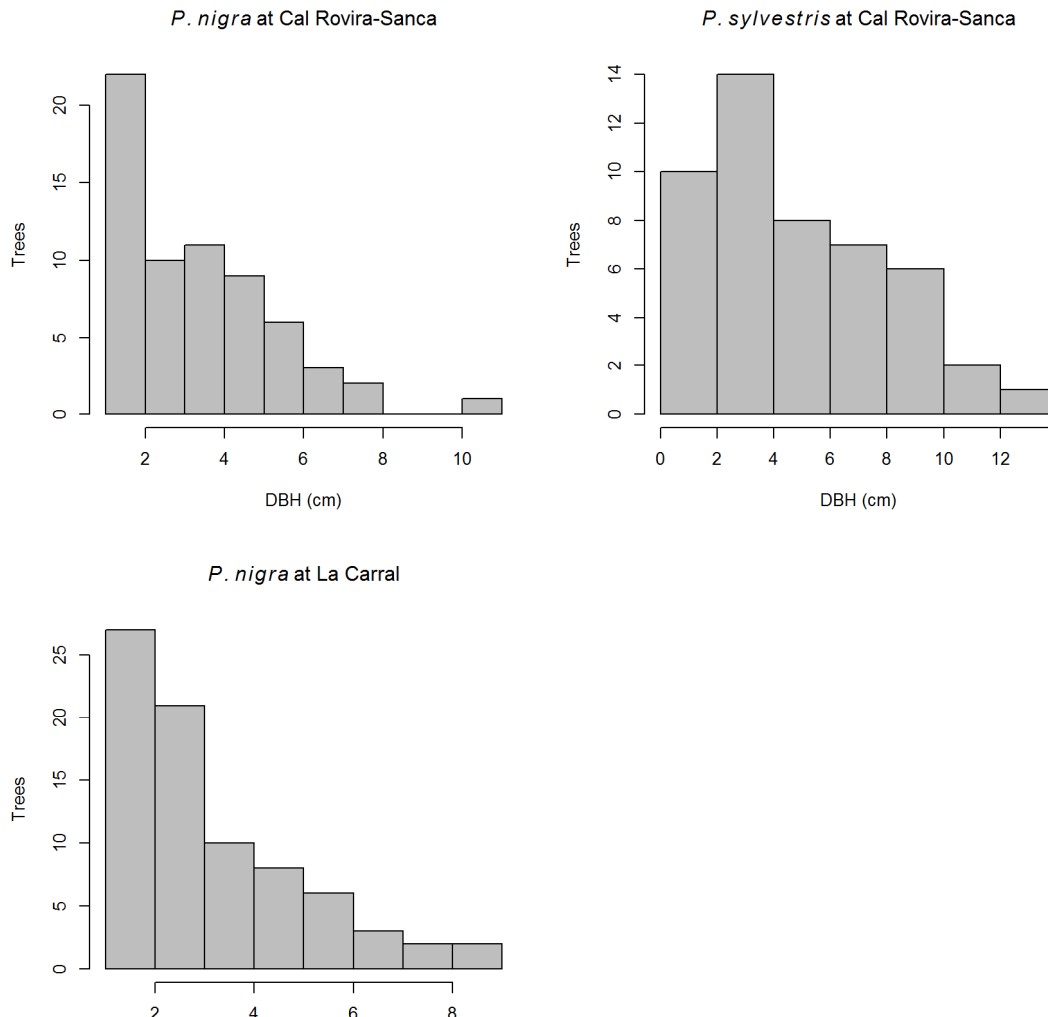

**Figure 3.** Distribution of DBH for all individuals of *Pinus nigra* and *Pinus sylvestris* found within the study area, both at Cal Rovira-Sanca and La Carral.

## 3. Results

Two pine species were found in both study sites, black pine (*Pinus nigra*) and Scots pine (*Pinus sylvestris*), although only one individual of the latter was recorded in La Carral. Mean tree height was 124.94 cm (s.d. = 88.78) for black pine and 188.15 (s.d. = 117.87) for Scots pine in Cal Rovira-Sanca and 109.82 (s.d. = 51.38) for black pine in La Carral, while the only Scots pine in this site was 230 cm high. DBH values were lower for black pine than for Scots pine, around 3.25 cm (mean = 3.36 and s.d. = 2.1 for Cal Rovira-Sanca; mean = 3.2 and s.d. = 1.87 for La Carral) versus 5 (mean = 4.91 and s.d. = 3.12 in Cal Rovira-Sanca; 6.6 cm for the only tree in La Carral).

The proportion of small trees was high for both species, but higher for black pine: median height of 117.5 and 105 and median DBH of 3.1 and 2.4 for Cal Rovira-Sanca and La Carral, respectively, as opposed to a median height of 192.5 and median DBH of 4.1 in Cal Rovira-Sanca for the Scots pine (see Figure 3 and Figure S3).

Due to flying times, both at La Carral and Cal Rovira-Sanca, light availability strongly limited the automatic selection of shutter speed, which resulted in a high motion blur (Table 1). Still, it was much higher in La Carral than in Cal Rovira-Sanca, which translated into additional distortion and reprojection errors (Table 1, Figure S4). Reprojection error was more than four times larger in La Carral than in Cal Rovira-Sanca and radial distortion much more exaggerated.

Greenness indices were revealed to be much better proxies for the sum of pine DBH than canopy height in both La Carral and Cal Rovira-Sanca. In fact, mean $R^2$ for the regressions of sum of pine DBH on the different statistics of greenness indices ranged from 0.005 to 0.466, while the regressions on canopy statistics ranged from 0.008 to 0.028 (Table 1; Figure 4). However, large differences were found among greenness indices (Figure S5-1). Even if GRVI and VARI still resulted in higher $R^2$ than any pine canopy height model, they were clearly outperformed by ExGI and GCC, which reached a maximum $R^2$ of more than 45% (Table 2).

Different statistics for the four greenness indices showed markedly differing fits, as shown by the high coefficients of variation of $R^2$ (Table 1; Figure 4). There were no clear-cut patterns of precision among the tested statistics that could be considered general for all four indices. Nevertheless, the count of non-zero values appears as the most consistently unreliable measure, with CVs higher than 100% for the all four indices. The best fitting model for one flight was not the same as for the others (Tables S6), which precludes considering any of them as a general best fitting model (Figure S7).

**Table 2.** Summary statistics of the determination coefficient ($R^2$) for the simple regressions of the sum of pine DBH on the four greenness indices and the canopy height model. Figures show the average $R^2$ across flights (two sites at each of two flight heights) and its coefficient of variation, expressed as percentage (within brackets). CHM stands for canopy height model. See "Imagery analysis" for a definition of the four indices.

|  | ExGI | GRVI | GCC | VARI | All Indices | CHM |
|---|---|---|---|---|---|---|
| Count of non-zero index | 0.154 (124.0) | 0.114 (118.4) | 0.147 (127.9) | 0.121 (119.8) | 0.134 (14.5) | 0.028 (50.0) |
| Max index value | 0.389 (27.2) | 0.164 (98.2) | 0.297 (37.7) | 0.051 (119.6) | 0.225 (65.9) | 0.014 (121.4) |
| Mean index value | 0.401 (12.7) | 0.195 (61.0) | 0.370 (50.5) | 0.039 (184.6) | 0.251 (66.9) | 0.015 (113.3) |
| Median index value | 0.227 (80.6) | 0.162 (50.6) | 0.182 (103.3) | 0.076 (118.4) | 0.162 (39.1) | 0.008 (125.0) |
| Std index values | 0.440 (37.3) | 0.152 (57.9) | 0.466 (32.2) | 0.005 (40.0) | 0.266 (84.5) | 0.018 (100.0) |
| Sum of index values | 0.256 (40.2) | 0.088 (78.4) | 0.155 (122.6) | 0.027 (122.2) | 0.132 (74.6) | 0.014 (114.3) |
| All measures | 0.311 (36.8) | 0.146 (26.4) | 0.270 (48.4) | 0.053 (76.8) |  | 0.016 (41.1) |

Canopy height alone shows a very poor explanatory power of the sum of DBH of pines, with a pooled mean of $R^2$ of 0.016, well below the values obtained from the four greenness indices. Accordingly, jointly considering greenness indices and canopy height does not increase adjusted $R^2$ more than 10% (Figure 4). For both sites and both flight heights, however, the best fitting model included always a canopy height statistics (Figure 5; Tables S6).

Different statistics of the canopy height model resulted in markedly different models (Table 1), with adjusted $R^2$s varying between 0.8 and 28%. The canopy statistics that provided the highest $R^2$ was the count of cells with non-zero values, although it was not the one producing the best models when combined with greenness indices (Table S1).

Flight height did not severely affect the capacity to estimate the sum of pine DBH per grid-cell (Figures 4 and 5). However, the best fits were always achieved for 120 m flights (Figure 4), although with a larger difference in Cal Rovira-Sanca.

Overall, the model with a best fit for the four flights included the standard deviation of the GCC index and the median of the canopy height model. It generally achieves fits close to those of the best fitting model for each of the flights (Figure 4 and Figure S5-1, Tables S3), although the difference was higher for the 50 m flight in Cal Rovira-Sanca.

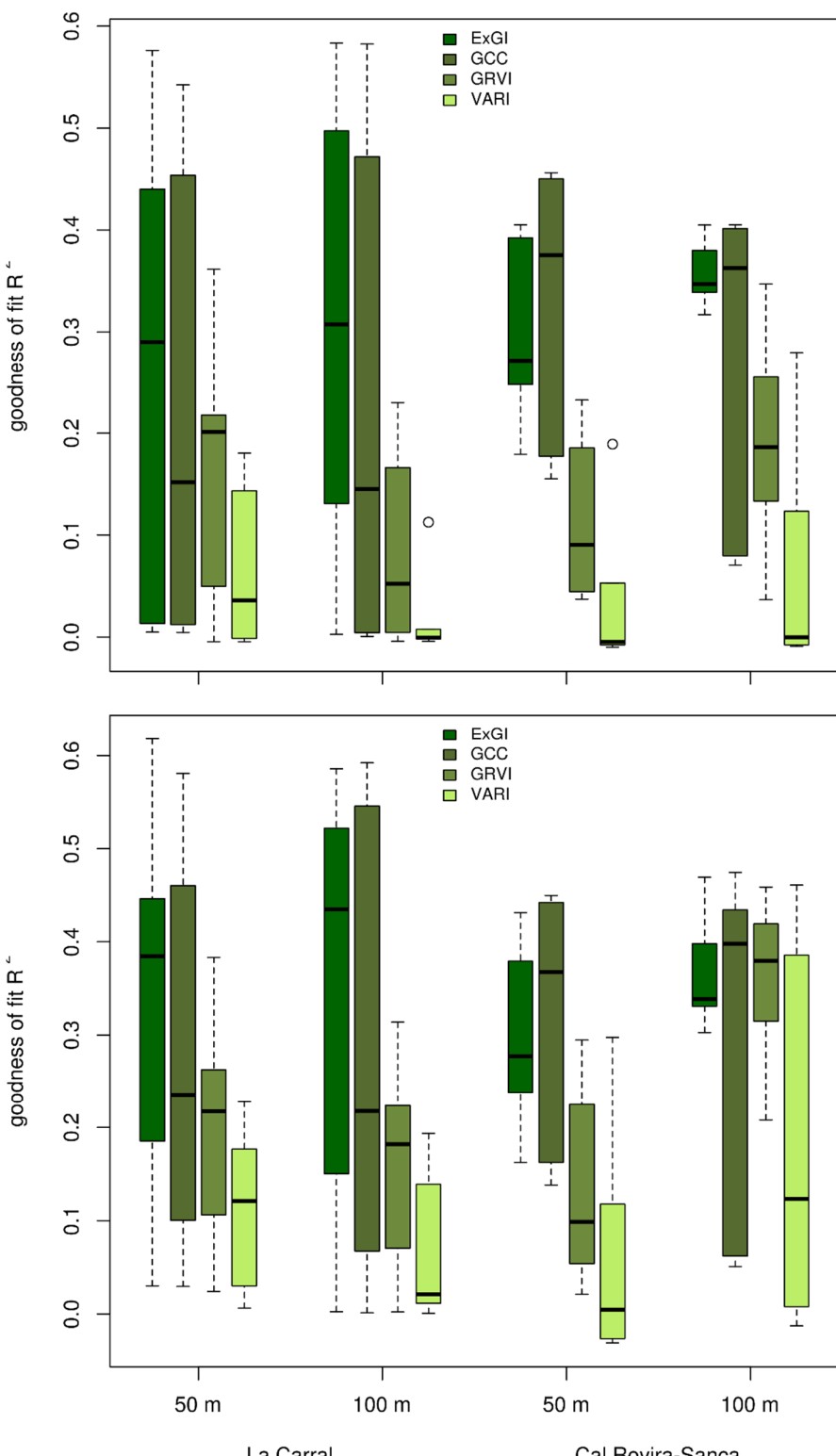

**Figure 4.** Values of the determination coefficient from the OLS regressions of the four greenness indices on the sum of DBH. Upper panel: simple regressions. Lower panel: multiple regressions where canopy height statistics were included as additional explanatory variables.

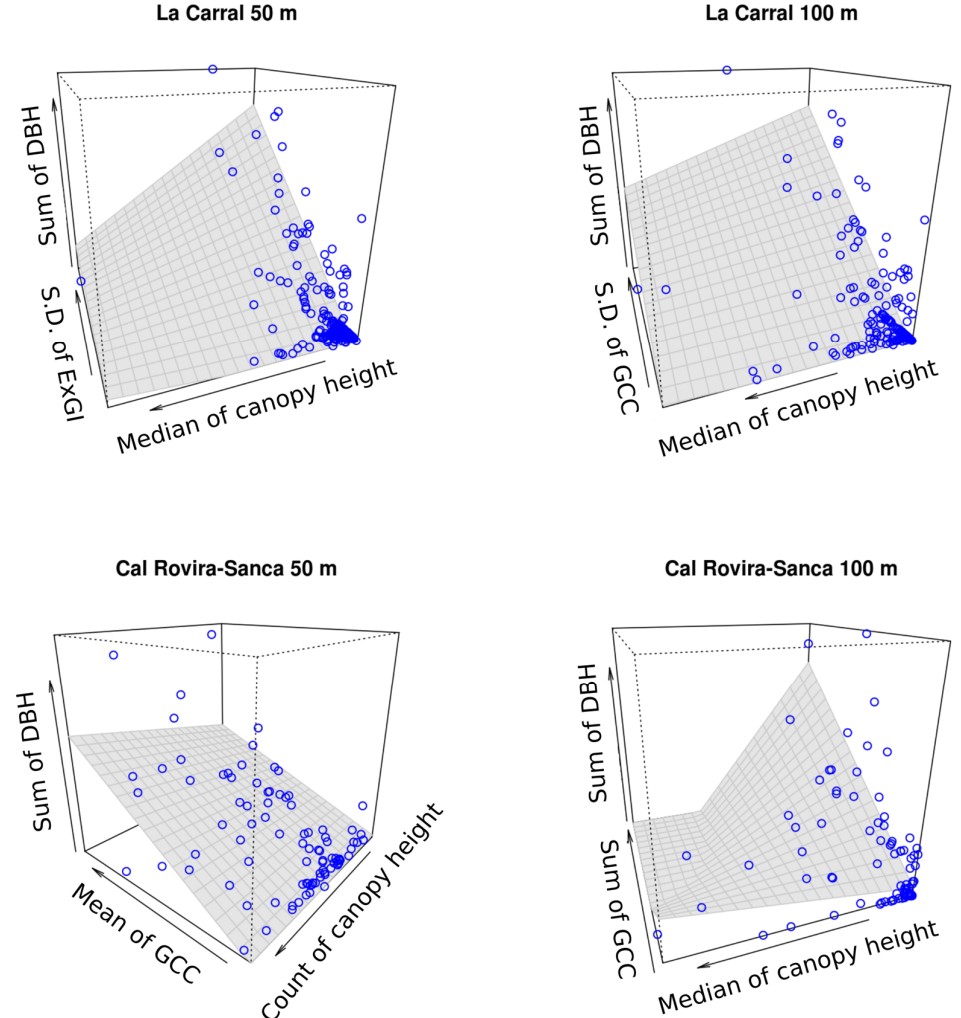

**Figure 5.** 3D scatterplots showing the best models. Each panel shows the bivariate regression that obtained the best fit (highest $R^2$) at each of the four flights. Blue dots indicate observed values, while the grey cover shows the fitted bivariate regression surface. Sum of DBH refers to the sum of the DBH of all the trees in each of the 5 m cells in which the study area was divided. Mean, s.d. and sum of GCC refers to the mean, standard deviation and sum of the Green Chromatic Coordinate values for all points of the point cloud within those same cells. S.d. of ExGi refers to the standard deviation of the Excess Green Index values for these same cells. Finally, median of canopy height model refers to the median value of the canopy height model (digital surface model) in each of the 5 m cells.

## 4. Discussion

UAVs are being increasingly used in ecological and conservation studies and monitoring, due to their multiple advantages, such as an increased spatial resolution, lower cost, higher acquisition flexibility and higher temporal resolution [1,2]. Low cost UAVs increase flexibility of use and reduce general costs as compared to professional platforms, and at the same time considerably reduce their learning slope, which has been deemed too steep. Yet, despite their low cost and ease of use, they accomplish spatial resolution way higher than those of aerial orthoimagery and provide useful results with a limited effort in image post-processing and analysis. This cost reduction is especially important for ecological studies and monitoring in non-profit forests and other habitats. In fact, nowadays, close forests make up only a small fraction of the experiments with UAVs in the field, while high value crops and forests are the main focus of most of them [6].

The flexibility and low cost of UAV deployment have been key factors determining the results of our pine estimation, as they allowed flying during the end of winter, before Portuguese oaks shed their leaves and the grass begins growing after the winter. Seeking the best moment to maximize spectral contrast among habitat components is paramount when it comes to discriminating different plant species or vegetation types with visual spectrum cameras [40].

Since the proposal of the first two vegetation indices by Pearson and Miller [57] and that of the NDVI shortly after by Rouse et al. [34], a high number of vegetation indices have been proposed and used for the study of vegetation from satellite, airplane and UAV sensor imagery (see for example references [32,38,39,41,58,59]). Most of them rely on the use of near infrared region (NIR), which profits from the marked difference in the absorbance spectrum of chlorophyll between red and near-infrared regions. The use and development of those indices that rely exclusively on the visual spectrum has sharply increased during last years, though, as a way to overcome the limited choice of spectrum bands in most UAV carried sensors. Assessment of their differential performance has yielded inconclusive results [31,32,35–37,39,41], but may even be better than those that include a NIR band [33,36]. Our results match those of Woebbecke et al. [31], who reported ExGI and GCC to perform better than other four indices when discriminating vegetation from non-plant background in different light conditions. Sonnentag et al. [32] recommended using GCC over ExGI to examine vegetation phenology in repeat photography studies, due to its lower sensitivity to changes in scene illumination. In our two sites, GCC also achieved higher accuracies when estimating pine cover in regenerating forests than ExGI.

By combining these indices with information derived from a canopy height model, we obtained determination coefficients for the pooled DBH of pines of up to 60%. Similar studies in agricultural plots, relating vegetation indices with vegetation fraction or cover [7,33,60], and in forest studies [8,61] report comparable figures. For example, Puliti et al. [8] achieved a 60% determination coefficient when estimating basal area from UAV imagery by combining point density and height. In our case, the canopy height model only improved the fit slightly (around 10%), probably due to the small size of most of the pines in both of our study sites. Later stages in the regeneration of vegetation would probably change the role of canopy height models in pine abundance prediction, as they would better allow discriminating pines from tall grasses and shrubs. Trials with canopy height of green pixels did not improve the fit of our models in this early stage (unpublished data).

Automatic exposition mode produced contrasting shutter speed values between sites, which translated into very diverse motion blur and radial distortion values. Surprisingly, we attained better results in La Carral, where motion blur was much larger than in Cal Rovira-Sanca. This larger motion blur resulted also in a marked radial distortion. Motion blur reduces the number of detected feature points and hence affects the quality of the photogrammetric processes [62]. A motion blur of two pixels has been deemed as enough to alter the results of these processes [63], although more recent work challenges this claim [62]. We estimated motion blur values below this threshold for the four flights, which could help explain the lack of relationship between motion blur and model fit performance at both sites. There were more evergreen shrubs in La Carral (eight shrubs) than in Cal Rovira-Sanca (one shrub) and the grid was built to avoid the green field at the eastern margin of Cal Rovira-Sanca area. Hence, we can rule out the presence of shrubs or other green vegetation as the reason for the lower performance of regressions in Cal Rovira-Sanca. The images in La Carral were taken with a much higher side and forward overlap, which could underlie the larger determination coefficients obtained in this site. Alternatively, the difference between flying times could have produced the difference between both sites. Lower sun angles result in changes in color temperature of the downwelling illumination [64,65], which would result in higher contrast between red and green channels.

We did not find marked differences between two contrasting flight altitudes in the general performance of greenness indices, although the selection of optimal greenness index or cell statistic depended on altitude at both sites. Flight height was an important determinant of the estimate of vegetation cover of wheat and barley experimental crops, together with growth stage and stitching

software [33]. Flight height determines the final image resolution obtained as well as the effect of topography on radiance (by changing the relative angle between terrain slopes and the UAV) and hence, values of greenness indices [33,66]. A change of scale can be expected to modify the determination coefficient of greenness indices, by changing the greenness values of each pixel [67]. However, we did not notice any remarkable effect, probably because our analysis relied on descriptive statistics encapsulating information at a coarser spatial extent of 5*5 m (Figure S5). Topography was not expected to introduce important variations on the values of greenness indices, as it was relatively mild and homogeneous in our filed plots. Actually, in contrast with our expectations, at both study sites 120 m flights resulted in higher determination coefficients than 50 m ones. Given the differences among flights, in our case either higher overlap values or lower proportional motion blur could be the drivers of this trend. However, the obtained regression fits are much better in La Carral than in Cal Rovira-Sanca, despite their much higher motion blur. Hence, our results suggest that the key parameter determining the determination coefficient of sum of DBH on greenness indices is a higher image overlap, rather than motion blur.

The UAV deployment and imagery analysis we present here has its own limitations, leading to areas for further improvement of image acquisition and analysis. First, the camera used has a FOV of 110° and non-rectilinear lenses, which results in a heavy fish-eye effect and a significant barrel distortion. Although the processing of images within Agisoft Photoscan corrects for lens distortion before carrying out the image matching process, the effect of distortion on image quality is too high toward the edges of the image and may result in a low quality in important fractions of the orthoimage. Correcting the geometric distortion of images before loading them in the photogrammetric software could improve accuracy and geometry of the final orthomosaic. This correction can be carried out in specialized software and should also consider the distortion caused by the electronic rolling shutter and motion blur resulting from aircraft movement [23,62]. Newer UAVs harbor improved quality cameras, even at the low cost range of products, with rectilinear lenses and more limited geometric distortions, that would help improve the detection of photosynthetically active vegetation.

Second, trying to reduce complexity of deployment and analysis to a minimum, we did not place ground control points. Spatial accuracy of the obtained orthoimage was thus limited by the resolution of the PNOA imagery (pixel size of 25 cm and RMSE below 50 cm) [68] and by the error associated to the identification of common points in the obtained orthoimage and in the PNOA orthophotograph. Although most present day low cost UAVs geotag the images taken on the fly, their GPS accuracy is still low and commonly results in RMSEs above 1 meter [20]. Hence, if we aim to reduce costs to a minimum, we need to deploy several ground control points (GCP) around the plot to increase the spatial accuracy. The spatial accuracy we get when using GCPs will ultimately depend on the accuracy of GCP location measurement, which may be reduced to centimeters or even millimeters with differential (DGPS) or real time kinetic (RTK) GPS measurements [20]. A software-as-a-service (SaaS) can also improve accuracy at an affordable cost, allowing RMSE values around four times lower than those from built-in GPS devices of consumer-level UAVs [20].

Third, the camera we used records data only on the visual spectrum. While this is one of the strengths of our work (due to its availability and ease of operation), it also limits the capacity to properly discriminate the photosynthetically active vegetation from the dormant vegetation. Capturing near infrared radiation (NIR) requires a more expensive, multispectral sensor. These kind of multispectral cameras are now more readily available for UAVs deployment. The assessment of its value in forestry is under current development, but they will probably prove especially suitable for multitemporal image acquisition and comparison [13,69,70].

## 5. Conclusion

Our results show that low cost UAV platforms are a useful alternative to professional platforms when it comes to the detailed, cost-effective monitoring of forest ecosystems and forest recovery after disturbance, especially in non-profit forests where economic and human resources may be scarcer.

By using a simple approach with a consumer level UAV platform and low cost (educational license at 179$) or free software, we have predicted post-fire pine recovery (estimated as the sum of pine DBH) with a relatively high coefficient of determination of up to 60%. Furthermore, all the software used has been applied with its default values, avoiding tweaking of any of their parameters. This allows the use of these types of procedures by end users working in forestry and applied research, and at the same time leaves margin for further improving the accuracy of the process. All in all, consumer level UAVs can be expected to provide a common low cost tool for ecological monitoring of post-fire recovery and other conservation and monitoring tasks, given future drops in prices, increasing accuracy levels and widening in application types.

**Supplementary Materials:** The following are available online at http://www.mdpi.com/2504-446X/3/1/6/s1, Figure S1.1. A detail of a high-cover area within the Cal Rovira-Sanca area, Figure S1.2. A detail of a low-cover area within the La Carral area, Figure S2-1. General workflow followed for the analysis of UAV imagery and the software frameworks used in each of the steps. Numbers refer to the different intermediate outputs shown in figure S2, Figure S3-1. Height distribution of *Pinus nigra* and *Pinus sylvestris* in La Carral and Cal Rovira-Sanca, as measured in the field, Figure S3-2. Regressions of tree height on DBH for both species in La Carral and Cal Rovira-Sanca, Figure S4-1. Image overlap obtained for the four flights. Figures depict the number of images where each point is present, Figure S4-2. Image residuals for the Phantom Vision FC200 sensor after camera calibration performed within Agisoft PhotoScan software. Scale of residual lines is indicated in pixel units, Figure S5-1. Maps of the four greenness indices and the estimated canopy model from the four flights carried out at both sites and two flight altitude, Table S6-1. Estimates of the best fitting model for the 50 m high flight in La Carral, with one greenness index (ExGI) and the pine canopy height model (CHM), Table S6-2. Estimates of the best fitting model for the 120 m high flight in La Carral, with one greenness index (GCC) and the pine canopy height model (CHM), Table S6-3. Estimates of the best fitting model for the 50 m high flight in Cal Rovira-Sanca, with one greenness index (GCC) and the pine canopy height model (CHM), Table S6-4. Estimates of the best fitting model for the 50 m high flight in Cal Rovira-Sanca, with the greenness index (GCC) and one pine canopy height model (CHM), Figure S7-1. 3D scatterplots showing the overall best fitting model applied to each site and flight altitude.

**Author Contributions:** Conceptualization, A.R.L. and L.B.; methodology, A.R.L. and L.B.; software, A.R.L.; validation, A.R.L. and L.B.; formal analysis, A.R.L.; investigation, A.R.L. and L.B.; resources, L.B.; data curation, A.R.L.; writing—original draft preparation, A.R.L.; writing—review and editing, A.R.L. and L.B.; visualization, A.R.L.; supervision, L.B.; project administration, L.B.; funding acquisition, L.B.

**Funding:** ARL benefited from the support of the NEWFOREST Marie Curie IRSES project. Grant No. IRSES-GA-2013-612645.

**Acknowledgments:** We are indebted to Jan and Martí Brotons for their collaboration on field measurements.

**Conflicts of Interest:** The authors declare no conflict of interest.

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
