# Peer review of "Greenness Indices from a Low-Cost UAV Imagery as Tools for Monitoring Post-Fire Forest Recovery"

_drones, doi:10.3390/drones3010006_

Round 1
Reviewer 1 Report
I believe the topic is interesting and the whole idea, methodology and results have been explained efficiently and appropriately. There are some part which should be explained more
- Line 69: 18 years?
- If there is any reason for selecting the applied indicators
- The software used for evaluation of accuracy
- The used tools and method for measuring tree parameters and their error
- Quality of figure 3 should be increased
Author Response
Comments and suggestions for authors
I believe the topic is interesting and the whole idea, methodology and results have been explained efficiently and appropriately. There are some part which should be explained more
- Line 69: 18 years?
Corrected to 20 years.
- If there is any reason for selecting the applied indicators
Yes, among the plethora of vegetation indices we selected these particular ones firts of all because they all can be calculated from a visual spectrum and for particular reasons (ExGI is very widely used and it has been proved to work, GCC improves it by reducing its sensitivenss to changes in scene illumination, GRVI correlates with NDVI when monitoring leave senescence and VARI is intended to improve it by accounting for atmospheric effects). We have now explained these issues in more detail.
- The software used for evaluation of accuracy
We did not measure accuracy, as we did not use GCPs.
- The used tools and method for measuring tree parameters and their error
We have now included a couple of sentences providing additional details on field sampling.
- Quality of figure 3 should be increased
We agree, in fact there was an error during the export process in the previous version. A high quality figure is included now.

Reviewer 2 Report
Overall comments:
- In overall, the study presents a valid application of UAV. Its presentation in its recent state, however, is too weak for a publication.
- In general, the reader can follow the ideas of the authors
- The methodical approach holds some shortcomings, which at least need to be thoroughly discussed (see below, details).
- Overall remark: it is more important to understand which processing algorithms have been used instead of which software was applied.
- Outline the advantages of UAVs for this task more clearly.
- There is no conclusions/summary
- The quality of the English is not good enough. This includes typos, missing words, missing blanks, deficiencies in expression etc. There are too many flaws for a review process.
Detailed comments:
- Page 2, lines 50-57: Was this section written by the same authors?
- Page 2, line 75: missing word in sentence
- Page 2, line 77: missing comma before “and”
- Page 2, line 79: flights
- Page 2, line 84: What are optimal weather conditions?
- Page 2, line 84: What is the rationale for flying two altitudes? Altitude over terrain or over tree tops? What about the image overlap? This should change if you acquire images every two seconds. What is the flight speed? What about motion blur? What about the radiometric quality of the sensor?
- Page 2, lines 84-89: What is the reason for using “-“ this often? This syntax makes no sense.
- It would be helpful to provide ground based photographs from the test site.
- Provide exact times of acquisition and the sun angle.
- Page 2, line 91: missing word
- Page 2, line 92: blank between number and unit “1 m”
- Why dbh at 1 m height? Most allometric models expect measurements at 1.3 m height.
- I stop here to correct this kind of flaws. Please check the whole paper and correct English usage.
- Page 2, line 102: As not everybody is aware about the functions implemented in XnViewMP, please describe the algorithms that have been applied
- Page 3, line 116ff: How do you derive terrain height from aerial images?
- Fig. 3: Orthoimages never show terrain, they are “flat”
- Why didn’t you use Photoscan for clipping and cleaning points?
- Page 3, line 136ff: Commonly these indices require calibrated data. If I understood correctly, you did no absolute calibration (which in fact is possible using reference targets). You need to discuss how this impacts your work and the comparability with similar studies.
- What is the impact of jpeg compression on the indices?
- Page 5, line 184: The sum of dbh is correlated with tree density. Discuss this issue in more details, otherwise the reader does not see the rationale for using the dbh sum. How is canopy cover related to dhb sum? If tree height is not related to canopy cover it is not surprising that tree height is not correlated with the greenness indices.
- Is a cell size of 5x5 m² sufficient to cover a wide range of dissimilarities on the one hand and representative samples on the other hand? It might happen that there is no tree within one cell.
- Surprising is the lower performance for the data with the higher resolution. How do you explain this?
- Fig.4: Difficult to interpret. Provide a helping figure caption please.
- Page 9, line 260ff: To show results for both flight altitudes is not an analysis.
- Page 9, line 260: Monitoring means regular observations – this is not covered in your study
- Page 9, line 260-292: Reads like a summary/conclusion, not a discussion
- Page 9, line 271: please mention the price for Agisoft/Photoscan
- Page 11, lines 330-345: Please discus more thoroughly, provide references when provide figures or qualitative statements that go beyond your analysis
- Page 11, lines 355: Higher accuracies than what?
No conclusions section?
Author Response
Overall comments:
- In overall, the study presents a valid application of UAV. Its presentation in its recent state, however, is too weak for a publication.
- In general, the reader can follow the ideas of the authors
- The methodical approach holds some shortcomings, which at least need to be thoroughly discussed (see below, details).
Thanks to the suggestions of reviewers of providing far more details on the quality of the resulting orthomosaics we feel the paper has really improved in our new version, as it has allowed us to analyze our results with a greater insight. See below for detailed responses to all the comments.
- Overall remark: it is more important to understand which processing algorithms have been used instead of which software was applied.
Our approach was to make the workflow easily available for the end user, which would probably not be aware of algorithms. Unfortunately, we did not find information on the algorithms used by XnView for automatic level and contrasts ajustments and Agisoft has not either released the details on the algoritms they use (see for example http://www.agisoft.com/forum/index.php?topic=89.0).
- Outline the advantages of UAVs for this task more clearly.
We think they were already included in the introduction, but we agree they were not clearly outlined. We have now done so at the end of the introduction.
- There is no conclusions/summary
We have now included a conclusion section, despite not being mandatory in Drones.
- The quality of the English is not good enough. This includes typos, missing words, missing blanks, deficiencies in expression etc. There are too many flaws for a review process.
We have now thouroughly review the typos, missing words, blanks, etc., and have improved the English writing.
Detailed comments:
- Page 2, lines 50-57: Was this section written by the same authors?
We are not sure to understand the point here. As stated in the contribution section, all the text has been originally written by A.R.L. and further edited and reviewed by both authors, A.R.L. and L. B. This, of course, includes the outlined paragraph.
- Page 2, line 75: missing word in sentence
Corrected.
- Page 2, line 77: missing comma before “and”
Right, we’ve corrected it now.
- Page 2, line 79: flights
Corrected.
- Page 2, line 84: What are optimal weather conditions?
We have now explicited the weather conditions we had for flying.
- Page 2, line 84: What is the rationale for flying two altitudes? Altitude over terrain or over tree tops? What about the image overlap? This should change if you acquire images every two seconds. What is the flight speed? What about motion blur? What about the radiometric quality of the sensor?
We flew at two altitudes with the aim of checking the effect of altitude in the capacity to detect number and size of pine trees in the area. We have now included this information, together with more detailed information on the parametres of the four flights, in the text. We have also corrected some error we had on our flight parameters. Unfortunately, we did not performed any radiometric calibration and don’t have any measure of the quality of this sensor.
- Page 2, lines 84-89: What is the reason for using “-“ this often? This syntax makes no sense.
We have erased the long dashes and changed the wording of the sentence. We hope now is clearer.
- It would be helpful to provide ground based photographs from the test site.
We have now included one picture of each of the study sites in the supplementary material.
- Provide exact times of acquisition and the sun angle.
We have now included these data in an additional table, describing the four flights, as suggested by reviewer 3.
- Page 2, line 91: missing word
We cannot identify the missing word in this sentence, but we have now re-worded the sentence in order to avoid the issue.
- Page 2, line 92: blank between number and unit “1 m”
Corrected.
- Why dbh at 1 m height? Most allometric models expect measurements at 1.3 m height.
We used DBH at 1 m for easiness, given the large proportion of small trees in the area. We have now explained that in the main text.
- I stop here to correct this kind of flaws. Please check the whole paper and correct English usage.
Yes, sorry for that. We had last moment problem with the template and could not check it again in detail. We have now combed the whole text for typos, missing words and other language corrections.
- Page 2, line 102: As not everybody is aware about the functions implemented in XnViewMP, please describe the algorithms that have been applied
Unfortunately, XnViewMP is not an open software project, and there is no reference in its documentation to the particular algorithm its function uses.
- Page 3, line 116ff: How do you derive terrain height from aerial images?
We forgot to mention that we took the altitudes for control points from the DEM provided by the Spanish Ministry (together with the PNOAA). We have now included that information.
- Fig. 3: Orthoimages never show terrain, they are “flat”
We are not sure to understand the point here. We guess the reviewer refers to the display of orthomosaics in figure 2, which are shown drapped on a DEM to easen the composition and visualization of the figure. We have now explained this issue in the legend.
- Why didn’t you use Photoscan for clipping and cleaning points?
For no particular reason, other than being used to CloudCompare for this tasks.
- Page 3, line 136ff: Commonly these indices require calibrated data. If I understood correctly, you did no absolute calibration (which in fact is possible using reference targets). You need to discuss how this impacts your work and the comparability with similar studies.
We have now included a short discussion on the limitations the absence of calibration imposes in our results as regards comparability among images and studies. We complete this discussion with examples where differences between green biomass and senescent leaves has been monitored even without calibration greenness indices.
- What is the impact of jpeg compression on the indices?
JPEG compression is a lossy compression method which prioritizes brightness over color, resulting in an important reduction of dynamic range of the picture and certain degree of image posterization (hard color changes and lower number of colors in the image). As a result, radiometric resolution decreases, which should reduce the ability to discriminate among different terrain or vegetation categories based on theis visual spectrum. However, except at the most extreme compression ratios (97%), JPEG compressed images still can successfully discriminate among different phenological stages of the vegetation (Sonnentag et al. 2012). We have now included this issue in the text.
- Page 5, line 184: The sum of dbh is correlated with tree density. Discuss this issue in more details, otherwise the reader does not see the rationale for using the dbh sum. How is canopy cover related to dhb sum? If tree height is not related to canopy cover it is not surprising that tree height is not correlated with the greenness indices.
The sum of dbh is certainly a combined measure of pine density and mean size and it has been shown to correlate with canpopy cover and biomass. Height has also been related to canopy cover and biomass. We provide now some details on the issue in the text.
- Is a cell size of 5x5 m² sufficient to cover a wide range of dissimilarities on the one hand and representative samples on the other hand? It might happen that there is no tree within one cell.
Sure, and certainly it happens. But those zeros are also included in our analyses and regressions. We think it is important to do so, given that the existence of gaps or areas with low cover is not a variable under control during an operational application of the workflow and hence including areas with and without trees is important when testing the adequacy of the approach. The case of post-fire monitoring is specially prone to the existence of a high proportion of cells without trees.
Hence, we still deem a 5 m cell as appropriate for the size of our study area, given that most of the pines within the area are still small and that increasing cell size would result in a exponential reduction in sample size for the statistical procedures.
- Surprising is the lower performance for the data with the higher resolution. How do you explain this?
It is surprising to us, also, and we are not certain of what can be the reason. We envisage two non-exclusive alternative, either higher overlap values or lower proportional motion blur improve the performance of our workflow. However, the obtained regression fits are much better in La Carral than in Cal Rovira-Sanca, despite their much higher motion blur. Hence, it seems that it is a higher overlap the key parameter determining the success of using greenness indices as a predictor of sumDBH, rather than motion blur.We have now included this issue in the discussion.
- Fig.4: Difficult to interpret. Provide a helping figure caption please.
We have expanded the figure caption to better explain its content. We hope it is now clearer. However, given the complexity of the explanation we are not certain it is clear enough. Please, let us know if it needs further clarification.
- Page 9, line 260ff: To show results for both flight altitudes is not an analysis.
We agree and have now changed analyzed to explored.
- Page 9, line 260: Monitoring means regular observations – this is not covered in your study
We changer monitor to assess.
- Page 9, line 260-292: Reads like a summary/conclusion, not a discussion
We agree and we have now erased the short summary paragraph, and moved the second on to the new conclusion section.
- Page 9, line 271: please mention the price for Agisoft/Photoscan
We have done so.
- Page 11, lines 330-345: Please discus more thoroughly, provide references when provide figures or qualitative statements that go beyond your analysis
We have improved the discussion, thanks to the suggested quality measures we have incorporated in the text, and we have included additional references on the issue.
- Page 11, lines 355: Higher accuracies than what?
We have now reworded the sentence to make it clearer.
No conclusions section?
Conclusion secton added.

Reviewer 3 Report
Dear authors,
ABSTRACT:
Lines 22-23: Could you please spell the abbreviations? Diameter at breast height (DBH) is a well-known abbreviation for the environmental community, Structure from Motion (SfM) is well known for the UAV community, but the vegetation indices abbreviations are not usually known for the readership. Therefore, and given that you have enough space in the abstract, I recommend you to spell all the abbreviations.
SECTION 1:
Line 33: Could you please spell DIY? I suppose you refer to "Do It Yourself"
Line 47: Mistake -> continuous,sly
Lines 50- 52: To support this idea regarding the relation between the spatial accuracy and the required investment, I suggest the following reference:
Padró JC, Muñoz FJ, Planas J, Pons X (2019). Comparison of four UAV georeferencing methods for environmental monitoring purposes focusing on the combined use with airborne and satellite remote sensing platforms. International Journal of Applied Earth Observation and Geoinformation, 79, 130-140. DOI: 10.1016/j.jag.2018.10.018. https://doi.org/10.1016/j.jag.2018.10.018. URL: https://www.sciencedirect.com/science/article/pii/S0303243418306421
SECTION 2.1:
Line 85: Could you please note the central hour of flight, or the beginning and the ending time of flight? it is an important feature to know the illumination geometry and the projected shadows. I recommend to add a table resuming the main features of each of the four flights: start time, final time, center of scene latitude and longitude, number of photograms and flight height, area covered, and, if you consider it appropriate, the number of pines. It will be good for the reader.
SECTION 2.2:
Line 105: What is the recommended workflow within Photoscan? Do you have any references? I suggest one of these if is what you used:
https://nctc.fws.gov/courses/references/tutorials/geospatial/CSP7304/documents/JeffSloan_2_000.pdf
https://nctc.fws.gov/courses/references/tutorials/geospatial/CSP7304/2016documents/HandsOn_Afternoon/UAS/UAS%20II%20Post%20Processing/PhotoScan%20Processing%20Procedures%20DSLR%20Feb%202016.pdf
Lines 126 and 128: I consider that referring the software [] is better than naming it, because the important is what you make, not the tool you use make it, since it is supposed that the science is repeatable and not proprietary. I suggest deleting explicit citations to particular software and just refer to them. In the case of Agisoft, you can maintain the name if you consider it appropriate, because the workflow is based in this photogrammetric software, but in the case of Fusion and CloudCompare you can write somehow "specific LIDAR tool for forest LIDAR data analysis [25]" and "they were cleaned up with a point cloud editing ...". The same idea is suggested for QGIS in line 170.
Line 127: I don't consider that "tool clip" has to appear in this paper. The same in the rest of the paragraph and tools.
Line 145: Here, I miss the abbreviaion GCC after "green chromatic coordinate".
Line 152: Here, I miss the abbreviaion GRVI after "green red vegetation index"
Line 180: Please, revise "5*5-m2".
SECTION 2.3:
Line 185: Missing space -> "assessthe"
Line 187: I find very interesting the Supplementary Materials S1 plots, because they relate field measurements and drone based estimations. Then, I suggest to move those figures to the 2.3 section.
SECTION 3:
Line 209: Check out the spaces before and after the = symbol.
Lines 204-210: If possible, I would like to see a boxplot showing the distribution of the pine heights, just to have a graphical idea of the pine height distribution. I think this would enrich the comprehension of the study area and your paper. In fact, I consider that this paragraph is not a result of your "Greenness indices from low-cost UAV imagery" research; I think that it is a description of the study area. However,
Lines 243-246: I consider that a subsection with three lines is not a correct way to structure the results. Please, join it with the previous or the next section, or expand the explanation.
Lines 255-258: I consider that a subsection with three lines is not a correct way to structure the results. Please, join it with the previous section, or expand the explanation.
SECTION 4:
Line 272: Missing space between “DBH)” and “with”.
Lines 337-345: This discussion can be related with the reference suggested in section 1.
There is not any conclusion section in this article.
Date of review: 05 December 2018.
Author Response
Dear authors,
ABSTRACT:
Lines 22-23: Could you please spell the abbreviations? Diameter at breast height (DBH) is a well-known abbreviation for the environmental community, Structure from Motion (SfM) is well known for the UAV community, but the vegetation indices abbreviations are not usually known for the readership. Therefore, and given that you have enough space in the abstract, I recommend you to spell all the abbreviations.
We agree, and now have spelled all abreviations in the abstract.
SECTION 1:
Line 33: Could you please spell DIY? I suppose you refer to "Do It Yourself"
Done.
Line 47: Mistake -> continuous,sly
Corrected.
Lines 50- 52: To support this idea regarding the relation between the spatial accuracy and the required investment, I suggest the following reference:
Padró JC, Muñoz FJ, Planas J, Pons X (2019). Comparison of four UAV georeferencing methods for environmental monitoring purposes focusing on the combined use with airborne and satellite remote sensing platforms. International Journal of Applied Earth Observation and Geoinformation, 79, 130-140. DOI: 10.1016/j.jag.2018.10.018. https://doi.org/10.1016/j.jag.2018.10.018. URL: https://www.sciencedirect.com/science/article/pii/S0303243418306421
We were aware of this recent paper. Thanks for pointing out to it. We have now included, as suggested.
SECTION 2.1:
Line 85: Could you please note the central hour of flight, or the beginning and the ending time of flight? it is an important feature to know the illumination geometry and the projected shadows. I recommend to add a table resuming the main features of each of the four flights: start time, final time, center of scene latitude and longitude, number of photograms and flight height, area covered, and, if you consider it appropriate, the number of pines. It will be good for the reader.
We have now included such a table, although we have included additionail flight details (overlap and motion blur) instead of data on the number of pines, as the latter corresponds only to a small part of the flight.
SECTION 2.2:
Line 105: What is the recommended workflow within Photoscan? Do you have any references? I suggest one of these if is what you used:
https://nctc.fws.gov/courses/references/tutorials/geospatial/CSP7304/documents/JeffSloan_2_000.pdf
https://nctc.fws.gov/courses/references/tutorials/geospatial/CSP7304/2016documents/HandsOn_Afternoon/UAS/UAS%20II%20Post%20Processing/PhotoScan%20Processing%20Procedures%20DSLR%20Feb%202016.pdf
We followed the workflow proposed by Agisoft itself in its user manual. We now provide details on this issue.
Lines 126 and 128: I consider that referring the software [] is better than naming it, because the important is what you make, not the tool you use make it, since it is supposed that the science is repeatable and not proprietary. I suggest deleting explicit citations to particular software and just refer to them. In the case of Agisoft, you can maintain the name if you consider it appropriate, because the workflow is based in this photogrammetric software, but in the case of Fusion and CloudCompare you can write somehow "specific LIDAR tool for forest LIDAR data analysis [25]" and "they were cleaned up with a point cloud editing ...". The same idea is suggested for QGIS in line 170.
We are not sure to agree, as one of our aims was to make the workflow clear and accessible to stakeholders and final users, which won’t in most cases be aware of algorithms and other details. However, we see your point and we don’t have strong feelings on the issue, as long as the software remains cited, and thus, we have changed the text as suggested. We have also moved Photoscan to the references, as we feel Fusion is as important as Photoscan for the workflow, and in fact, Photoscan is the only non-free software.
Line 127: I don't consider that "tool clip" has to appear in this paper. The same in the rest of the paragraph and tools.
We have removed all references to specific tools within Fusion.
Line 145: Here, I miss the abbreviaion GCC after "green chromatic coordinate".
Included.
Line 152: Here, I miss the abbreviaion GRVI after "green red vegetation index"
Included.
Line 180: Please, revise "5*5-m2".
Corrected.
SECTION 2.3:
Line 185: Missing space -> "assessthe"
Corrected.
Line 187: I find very interesting the Supplementary Materials S1 plots, because they relate field measurements and drone based estimations. Then, I suggest to move those figures to the 2.3 section.
We partially followed your suggestion by including only the histograms of DBH, the variable finally modelled. We decided not to include Height distribution and the relationship between both variables, as that would imply including three large figures, which seemed to many to us. Of course, should the editor consider that including all of them is not too much, we would be happy to do so.
SECTION 3:
Line 209: Check out the spaces before and after the = symbol.
Corrected.
Lines 204-210: If possible, I would like to see a boxplot showing the distribution of the pine heights, just to have a graphical idea of the pine height distribution. I think this would enrich the comprehension of the study area and your paper. In fact, I consider that this paragraph is not a result of your "Greenness indices from low-cost UAV imagery" research; I think that it is a description of the study area. However,
It is, actually, a description of our study area, but since it’s a result obtained from the described field work, we still think it fits here. Of course, should the editor think it should be move, we’d be happy to do so. We have not included the suggested boxplot, as the distribution of pine heights is already provided in the form of a histogram in the supplementary material (S1), and is already referenced above.
Lines 243-246: I consider that a subsection with three lines is not a correct way to structure the results. Please, join it with the previous or the next section, or expand the explanation.
We agree and we have joined all the sections now.
Lines 255-258: I consider that a subsection with three lines is not a correct way to structure the results. Please, join it with the previous section, or expand the explanation.
We agree and we have joined them now.
SECTION 4:
Line 272: Missing space between “DBH)” and “with”.
Corrected.
Lines 337-345: This discussion can be related with the reference suggested in section 1.
Again, thanks for pointing out to this reference we were unaware of. We really find it useful and have now used for deepening our discussion.
There is not any conclusion section in this article.
We have now included a conclusion section, despite not being mandatory in Drones.

Round 2
Reviewer 3 Report
Dear authors,
Now, the article has a better comprehension and I agree on most of your answers. Then, I suggest minor revisions to improve the quality of the paper in some details.
ABSTRACT:
Lines 13 and 22: Don’t you think that the spelled abbreviations should begin with capital letters? i.e. Unmanned Aerial Vehicles (UAV) and Diameter Breast Height (DBH).
Line 24: Although they only appear once in the abstract, consider adding, for consistency, the abbreviation of Structure from Motion (SfM), Excess Green Index (EGI), Green Chromatic Coordinate (GCC), Visible Atmospherically Resistant Index (VARI) and Green Red Vegetation Index (GRVI).
SECTION 2.1:
The added table is very good on synthetizing the flights. However, there are some details in the first column that can be improved:
Date (DD/MM/YY)
Time (UTC)
Sun Zenith Angle (°) (Sun Azimuth Angle (°))
Center of Scene (UTM31N-ETRS89)
Flight height (m)
The Sun Zenith Angle is important to be explicitly mentioned because it can be confused with its complementary angle, the Sun Elevation Angle. Moreover, I suggest you to mode the Sun Azimuth Angle to a new table row to make it clearer.
SECTION 2.2:
You should revise the consistency on the use of capitals on spelling the abbreviations. For example, in line 143 you write “National Plan of Aerial Orthophography (Plan Nacional de Ortofotografía Aerea, PNOA)”, but in line 160 you write “digital terrain model (DTM)”. This happens in most of the abbreviations (lines 33, 103, 150, 160, 163-165…) and I suggest you to follow a rule where the words begin in capitals, followed by the corresponding abbreviation, as in the PNOA case.
Lines 199-207: Why “jpg” and “JPEG” are differently wrote? Is suggest to use the same name and in capitals. I also suggest to add (DN) after Digital Numbers in line 199, because in line 217 appear DN without a previous spelling.
Line 222: Consider revise “0For”.
SECTION 4:
Lines 421-422: In the reference [20] it is not tested any RTK, but maybe you refer to “…which may be reduced to centimeters with indirect georeferencing based on ground control points measured with differential GPS (DGPS), or with direct positioning using Post-Processed Kinematic (PPK) carrier-phase double-frequency GPS measurements [20]. The PPK single-frequency GPS measurements can also improve accuracy at an affordable cost, allowing RMSE…”.
SECTION 5:
Line 445: Please, fix double final point.
Date of review: 21 December 2018.
Author Response
ABSTRACT:
Lines 13 and 22: Don’t you think that the spelled abbreviations should begin with capital letters? i.e. Unmanned Aerial Vehicles (UAV) and Diameter Breast Height (DBH).
Although we also find it intuitive and useful, the correct usage in English (and the APS style manual) state that, despite acronyms and initialisms are written in capital letters, the words used to form them should not be capitalized, except when they are proper names or names of organims, instituitions and companies. We have herein followed this criterium.
Line 24: Although they only appear once in the abstract, consider adding, for consistency, the abbreviation of Structure from Motion (SfM), Excess Green Index (EGI), Green Chromatic Coordinate (GCC), Visible Atmospherically Resistant Index (VARI) and Green Red Vegetation Index (GRVI).
We have followed the common usage of written English and ASP style manual (see above) and have not capitalized those word, except for proper nouns.
SECTION 2.1:
The added table is very good on synthetizing the flights. However, there are some details in the first column that can be improved:
Date (DD/MM/YY)
Time (UTC)
Sun Zenith Angle (°) (Sun Azimuth Angle (°))
Center of Scene (UTM31N-ETRS89)
Flight height (m)
Changes applied.
The Sun Zenith Angle is important to be explicitly mentioned because it can be confused with its complementary angle, the Sun Elevation Angle. Moreover, I suggest you to mode the Sun Azimuth Angle to a new table row to make it clearer.
Actually, the angle provided is the sun elevation angle, which we have made explicit now.
SECTION 2.2:
You should revise the consistency on the use of capitals on spelling the abbreviations. For example, in line 143 you write “National Plan of Aerial Orthophography (Plan Nacional de Ortofotografía Aerea, PNOA)”, but in line 160 you write “digital terrain model (DTM)”. This happens in most of the abbreviations (lines 33, 103, 150, 160, 163-165…) and I suggest you to follow a rule where the words begin in capitals, followed by the corresponding abbreviation, as in the PNOA case.
We have now capitalized the words of acronym and initialism definitions exclusively when they correspond to proper names (see responses above).
Lines 199-207: Why “jpg” and “JPEG” are differently wrote? Is suggest to use the same name and in capitals. I also suggest to add (DN) after Digital Numbers in line 199, because in line 217 appear DN without a previous spelling.
We have now consistently used JPEG in capital letters, and have properly defined DN initialisms.
Line 222: Consider revise “0For”.
Corrected.
SECTION 4:
Lines 421-422: In the reference [20] it is not tested any RTK, but maybe you refer to “…which may be reduced to centimeters with indirect georeferencing based on ground control points measured with differential GPS (DGPS), or with direct positioning using Post-Processed Kinematic (PPK) carrier-phase double-frequency GPS measurements [20]. The PPK single-frequency GPS measurements can also improve accuracy at an affordable cost, allowing RMSE…”.
We thinks this comment origins from a misreading of the sentence. In this sentence we are talking on the precision of GCPs, which can be improved with DGPS or RTK. In reference 20, GCPs where georreferenced by means of static RTK. The next sentence is devoted to an affordable way to improve accuracy of the final 3D reconstruction result. We did not refere to the PPK2 option due to its high initial cost, which is much higher than the low-cost platforms we are exploring in this paper.
SECTION 5:
Line 445: Please, fix double final point.
Fixed.
